# Canadian Melanoma Conference Recommendations on High-Risk Melanoma Surveillance: A Report from the 14th Annual Canadian Melanoma Conference; Banff, Alberta; 20–22 February 2020

Christina W. Lee [1], J. Gregory McKinnon [2] and Noelle Davis [1,*]

1   Surgical Oncology, Department of Surgery, University of British Columbia, Vancouver, BC V5Z 1M9, Canada; christina.lee2@vch.ca
2   Surgical Oncology, Department of Surgery, University of Calgary, Calgary, AB T2N 2T9, Canada; Greg.Mckinnon@albertahealthservices.ca
*   Correspondence: ndavis@bccancer.bc.ca; Tel.: +1-604-875-5770; Fax: +1-604-875-4315

**Abstract:** *Introduction:* There are a lack of established guidelines for the surveillance of high-risk cutaneous melanoma patients following initial therapy. We describe a novel approach to the development of a national expert recommendation statement on high-risk melanoma surveillance (HRS). *Methods:* A consensus-based, live, online voting process was undertaken at the 13th and 14th annual Canadian Melanoma Conferences (CMC) to collect expert opinions relating to "who, what, where, and when" HRS should be conducted. Initial opinions were gathered via audience participation software and used as the basis for a second iterative questionnaire distributed online to attendees from the 13th CMC and to identified melanoma specialists from across Canada. A third questionnaire was disseminated in a similar fashion to conduct a final vote on HRS that could be implemented. *Results:* The majority of respondents from the first two iterative surveys agreed on stages IIB to IV as high risk. Surveillance should be conducted by an appropriate specialist, irrespective of association to a cancer centre. Frequency and modality of surveillance favoured biannual visits and Positron Emission Tomography Computed Tomography (PET/CT) with brain magnetic resonance imaging (MRI) among the systemic imaging modalities available. No consensus was initially reached regarding the frequency of systemic imaging and ultrasound of nodal basins (US). The third iterative survey resolved major areas of disagreement. A 5-year surveillance schedule was voted on with 92% of conference members in agreement. *Conclusion:* This final recommendation was established following 92% overall agreement among the 2020 CMC attendees.

**Keywords:** cutaneous melanoma; high risk; surveillance; recommendation statement

## 1. Purpose

The purpose of the Canadian Melanoma Conference Recommendation Statement is to define the best post-treatment surveillance practices for patients with high-risk cutaneous melanoma in Canada. A novel method of obtaining consensus of opinions among melanoma experts was used.

## 2. Participants

The CMC welcomes health care providers involved in the care of cutaneous melanoma ranging from surgical oncologists, medical oncologists, radiation oncologists, dermatologists, pathologists, and radiologists, and allied health professionals throughout Canada.

## 3. Target Audience

The conference statement presented here is targeted to health care professionals involved in the care of patients with cutaneous melanoma.

## 4. Basis of Recommendations

The recommendations put forth are based on the presentation of current evidence, discussion of challenging issues and expert informed opinions.

## 5. Introduction

Malignant melanoma is an aggressive disease with treatment that has radically improved over recent years. Novel targeted and immune-based therapies have led to improved patient survival [1,2]. Contemporary studies report disease recurrence rates to range from 30% to 47% among stages II and III melanoma [1,2]. Earlier detection of asymptomatic recurrence may improve opportunities for treatment and eligibility for clinical trials with additional impact on overall survival [1]. However, no consensus exists regarding an optimal surveillance strategy for high-risk patients.

There is an absence of a standard definition of what constitutes 'high-risk' in cutaneous melanoma. High-risk cohorts are broadly defined as those patients with worse overall survival at 5 years, including those with Stage IIB, IIC, IIIB, IIIC and IV according to the eighth edition of the American Joint Committee on Cancer TNM staging system (AJCC) [3]. A report from the United Kingdom identified nearly half (47%) of all patients deemed 'high-risk' (stages IIB-C, IIIA-C) had relapsed, and among these patients, 66% were asymptomatic [4]. Hence, imaging is necessary to detect low-volume disease in the majority of patients.

Although limited, there are data to support the role of surveillance imaging to identify clinically occult, asymptomatic melanoma recurrences [5]. There are conflicting data derived from small cohorts and retrospective series to suggest variable accuracies in the detection of melanoma metastases from various imaging modalities such as whole body 18F-fluoro-2-deoxy-D-glucose positron emission tomography computed tomography (FDG-PET CT), conventional CT, and US [3,4,6–8]. However, a survival benefit from early detection has yet to be demonstrated. The ideal surveillance strategy would implement re-evaluation of high-risk populations in a cost-effective manner with the goal of detecting subclinical disease relapse in order to initiate treatment with potential curative intent.

We sought to identify a high-risk group of patients who would benefit from surveillance. Secondly, we attempted to answer the following questions about the specifics of the proposed surveillance:

(a) Who should conduct high-risk surveillance?
(b) What form and modality of surveillance should be utilized in HRS?
(c) Where should HRS be conducted?
(d) When should HRS be conducted, and how frequently?
(e) How should HRS be conducted with respect to clinical exams, laboratory investigations and imaging modalities?

## 6. Methodology

A. A three-part, consensus-based voting process was designed to address issues related to surveillance of high-risk melanoma patients at the 2019, 13th annual Canadian Melanoma Conference (CMC) in Banff, Alberta, Canada. This served as the first iteration of questions about high-risk surveillance in patients with melanoma. The purpose of the survey was to gather expert opinions where evidence was lacking. All conference attendees were invited to participate in this live survey. A total of 26 experts contributed responses in this phase. A live, interactive voting tool (sli.do s.r.o.) was utilized to allow audience participants at the conference to submit answers to survey questions posed during a podium presentation. All results were anonymous and recorded live. Consensus was determined by majority vote for multi-tier answers, or >50% agreement in dichotomous answers. In order to clarify questions that failed to achieve a majority agreement, a second iteration questionnaire was developed.

B.    The second iteration of multiple-choice survey questions was disseminated via an online survey format. Participants were selected based on prior 2019 CMC attendance. In addition, recognized melanoma clinicians across Canada were invited to participate. Each participant was emailed a cover letter with a description of the project and link to the online survey. A total of 30 individuals participated in this second phase. The survey consisted of 10 multiple choice questions. Answers were formatted in either Likert scale or closed-ended options (Supplemental Materials A).

C.    The third iterative questionnaire was completed via live participation among conference attendees at the 14th annual Canadian Melanoma Conference in 2020, also in Banff, Alberta, Canada. The same live, interactive voting tool was used to collect responses to questions clarifying aspects of HRS pertaining to the "who, what, where, and when" surveillance should be conducted. All responses were anonymous. A total of 21 conference members took part in this third and final phase (Supplemental Materials B). Figure 1 depicts the methodological approach to achieving this consensus recommendation.

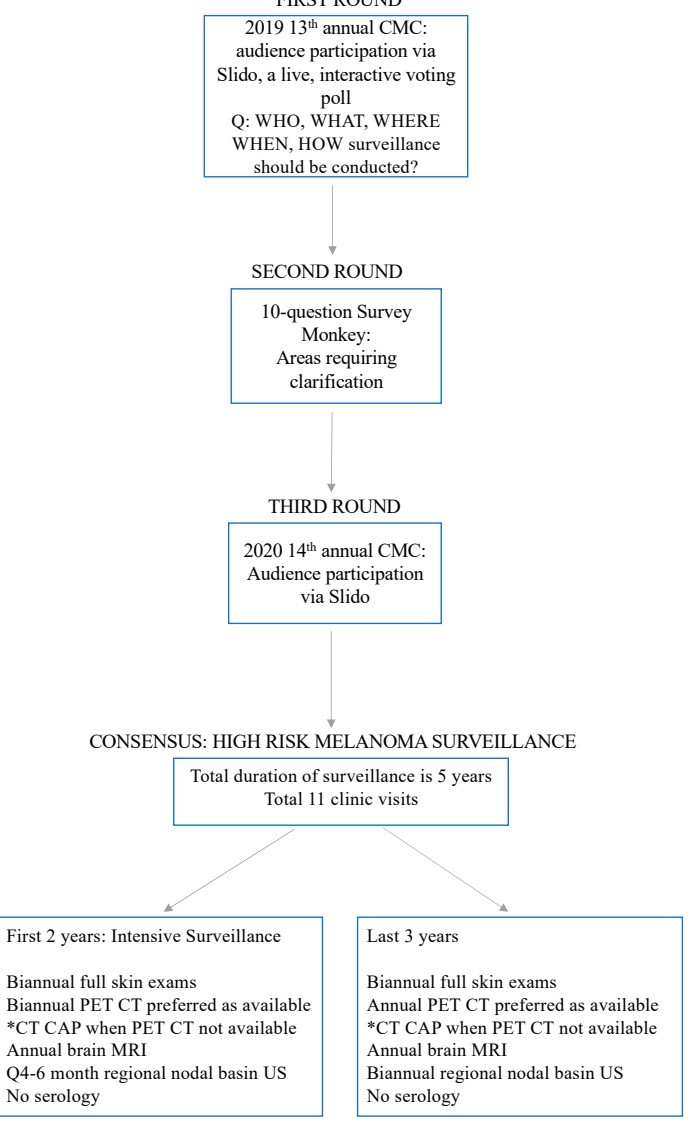

**Figure 1.** Methodological approach in achieving expert consensus in the development of recommendations on high-risk melanoma surveillance in Canada. * refers to the availability of PET-CT where care is delivered. PET-CT is recommended above conventional CT. Where PET-CT is not readily available, or limited by travel, conventional CT is acceptable.

### 7. Findings

Survey participants consisted of medical and surgical oncologists, dermatologists, radiologists and general practitioners. Majority responses were considered those with greater than 50% agreement.

In the first iteration questionnaire, questions were presented in a clinical case-based format in order to encourage discussion on selected answers. Consensus was reached regarding the "who and where" aspects of HRS. Respondents agreed on the inclusion of select stage II patients in the definition of "high-risk" and that a combination of melanoma specialists would be appropriate to conduct follow up at local or regional centers.

We identified areas lacking a minimum 50% majority among respondents from the first survey and developed clarifying questions for the second iteration questionnaire in the following domains: the definition of a high-risk population; the utility and frequency of PET-CT imaging versus usual cross-sectional imaging of the chest, abdomen and pelvis; and frequency of nodal basin ultrasonography. An intensive period of surveillance followed by routine surveillance was supported over a 5-year period. Systemic imaging was recommended every 6 months. CNS imaging was recommended on an annual basis. Regional nodal basin US was not recommended among sentinel lymph node-negative patients. Serum LDH was not supported in HRS. Consensus was not reached on the definition of "high-risk", optimal modality of systemic imaging (PET-CT versus conventional CT), and the impact of PET-CT utilization on frequency of regional nodal basin US during the intensive period of surveillance.

The third iteration of questions explored persistent areas of non-consensus, revisiting: which patients constitute high-risk, agreement of PET-CT as the preferred modality for systemic imaging and frequency of regional nodal basin US. Specifically, stage IIB to IV defined high-risk melanoma. The period of intensive surveillance would constitute the first two years after initial therapy. The frequency of regional nodal basin US (US) was further debated given the lack of data in support of surveillance protocols from recent clinical trials. Individual provider discretion was advised every 4 months versus 6 months during the intensive period. PET-CT was recommended above conventional CT where available. Patients in remote locations of Canada where conventional CT is available were not obligated to travel to reach centres with PET-CT availability.

#### 7.1. Definition of "High-Risk" Melanoma Patients

Pathologic stage migration occurred between the 7th and 8th editions of the AJCC staging and classification of cutaneous melanoma, evidenced by improved survival within similarly staged groups in the 8th edition. The expanded, four stage III subgroups demonstrated 5-year melanoma-specific survival (MSS) ranging from 32% to 93%, which are significantly better than the previously categorized 3 stage III subgroups in the 7th edition, ranging from 40% to 78% [9].

High-risk recurrence is defined by the AJCC stage, which include prognostic factors such as thickness and the presence of ulceration [9]. High-risk cohorts are broadly defined as those patients with worse overall survival (<50%) by 5 years, including those with Stage IIB, IIC, IIIB, IIIC and IV according to the eighth edition of the American Joint Committee on Cancer TNM staging system (AJCC) [3]. Additional reports support early recurrence patterns identifying nearly half (47%) of all patients deemed 'high-risk' (stages IIB-C, IIIA-C) had relapsed at a median time of 10.1 months, with 66% of these patients presenting with asymptomatic relapses [4]. Therefore, the consensus definition of high-risk patients was stages IIB–IV. There was strong agreement between the second and third questionnaires on this definition.

#### 7.2. Which Providers Should Be Conducting High-Risk Surveillance?

Expert participant specialists agreed that medical oncologists, surgical oncologists, dermatologists, and general practitioners with a special interest in cutaneous melanoma are appropriate to conduct high-risk surveillance.

Not all providers of varying general and/or subspecialized training may feel comfortable or willing to conduct HRS. Hence, the determination of a clearly defined surveillance schedule for resected stage IIB to IV patients was increasingly important in disseminating treatment algorithms to physicians who treat patients in a community setting. This was also intended to facilitate widespread standardization and may be used as a means of measuring the quality of care delivered.

### 7.3. Where Should Surveillance Be Conducted?

High-risk surveillance should be conducted at a facility where appropriate expertise and technological resources are available.

The conference acknowledged the limitations of health care resource availability across Canada, and the lack of access uniformity to all persons. In 2016, Statistics Canada published a report, "Difficulty Accessing Health Care Services in Canada", detailing that approximately 71% of Canadians over the age of 15 did not experience any difficulty with access to health care services. Among those who reported difficulty, these challenges were dependent on the services being sought, which focused on non-emergency surgery and selected diagnostic tests [10]. However, between 2003 and 2010, there was a significant decrease in patient-perceived difficulty for access to these services [10]. It was plausible to ascertain that for those in which healthcare resources are available, lengthy travel to a centralized, regional or provincial cancer center may be necessary for a select proportion of Canadians.

As the discussion unfolded regarding access and resource availability, the issue of PET-CT utilization across Canada was appropriately brought into question. The Canadian Institute for Health Information (CIH) contributes collected data on medical imaging technologies, which are further reported by the Canadian Agency for Drugs and Technologies in Health for the purposes of maintaining the Canadian Medical Imaging Inventory [11]. The most recent report published in 2018 unveiled a total of 51 PET-CT units across 45 sites available in Canada [11]. No PET-CTs were available in the territories and Prince Edward Island, whereas the greatest number of units was found in Quebec, Ontario and Alberta [11]. Eighty percent of PET-CT utilization in Canada was reported as oncology-related, and 52% of all units were being utilized less than 20 h per week [11].

Recognizing that a significant proportion of Canadians may experience difficulty accessing PET-CT and expert opinion in certain areas, 83% of those polled still agreed that high-risk surveillance would be best conducted regionally, so long as appropriate expertise and access to the necessary resources are available. Access to a cancer center via referral and/or consultation with a melanoma expert should be readily available to providers in distant and remote locations whenever feasible.

### 7.4. What Forms of Systemic Imaging Should Be Used for High-Risk Surveillance?
#### 7.4.1. CT of the Chest, Abdomen and Pelvis (CT C/A/P) and PET-CT

The recommendation favoured PET-CT over conventional CT, where available, for systemic imaging.

The recommendation for routine PET-CT remained controversial. This was evidenced by the lack of consensus throughout the second and third iteration surveys. It is well known that systemic imaging serves as an initial detection method for asymptomatic regional and distant relapses, as early reports have noted greater than 50% of first relapses occur systemically [12]. Prior investigations reported superior radiographic detection rates compared to conventional CT for every 8th edition AJCC stage, with pooled sensitivity and negative predictive value rates of 96% and 95%, respectively [3,6,13,14].

Our serial questionnaires asked for an expert opinion on the routine use of CT C/A/P versus PET-CT due to the absence of clear consensus. Various factors contributed to dissent such as the availability of PET-CT, remote geographic location and patient commute to access PET-CT imaging, availability of expert radiologists and cost. Fifty-five percent of respondents from the first survey iteration favoured PET-CT over CT C/A/P. The

second survey iteration similarly identified agreement in favour of PET-CT among 50% of respondents; however, 74% of respondents believed that given resource limitations, patients should not be required to travel far distances to undergo PET-CT if usual CT C/A/P was regionally available.

The use of FDG-PET CT or PET-CT deserves special attention in this recommendation. The overarching benefit of systemic imaging in surveillance is to identify subclinical relapses that cannot otherwise be identified on the basis of known prognostic parameters. FDG-PET CT has long been suspected to be of potential benefit, resulting in early series evaluating the use of routine FDG-PET CT in stage III melanoma after positive sentinel lymph node biopsy [13]. The added benefit of visualizing hypermetabolic tissues, along with the ability to detect functional lesions as small as 80 mm$^3$ in combination with CT, were reasons for PET-CT gaining popularity as a superior and non-invasive imaging modality in melanoma [8,15–17]. Earlier studies were limited by having a retrospective design and small cohorts, yielding low sensitivity and specificity ratios without convincing support for its routine utilization. Earlier studies report the sensitivity rate of FDG-PET CT to range from 67 to 100%, and specificity from 56 to 100% [8,15,17,18]. Further validation of the utility of PET-CT was supported by recent comparative studies among patients with stage III melanoma, where PET-CT demonstrated a superior 53% detection rate compared to physical exam alone, altering subsequent therapy options in approximately 50% of the patient cohort [13]. Further, Xing et al. identified superior sensitivity and specificity for detection of distant metastases with PET-CT compared to PET alone among stage II and III patients (Se: 86% vs. 82%, Sp: 91% vs. 83%; PET-CT vs. PET alone) [8].

### 7.4.2. Brain MRI

Expert recommendations unanimously identified annual brain MRI as the modality of choice in the detection of central nervous system relapse.

To date, there are no prospective investigations with long-term follow up on the detection of melanoma brain metastases. Brain metastases are associated with significant morbidity and often impart short median survival. Approximately 4% to 13% of recurrences occur in the brain in stage III and high-risk stage II patients [3,17,19]. A recent report utilizing the 2016 Surveillance, Epidemiology and End Results (SEER) Program identified the age-adjusted incidence of synchronous cutaneous melanoma brain metastases among individuals over the age of 65 derived from Medicare claims data between 2008 and 2012 to be 1.1% among 35,268 cases [20]. These results suggest cranial evaluation is an important aspect of HRS.

### 7.4.3. Regional Nodal Basin Ultrasound

Nodal basin ultrasound should be performed every 4 to 6 months during the first two years of intensive HRS. The final expert recommendation agreed on provider discretion in regard to the chosen frequency between 4 and 6 months based on an individual patient risk assessment.

All respondents from the first survey provided consensus on ultrasound imaging as the ideal modality for regional nodal basin surveillance in sentinel lymph node-positive patients. Only 50% of respondents from the second iteration agreed on US at every 4 months as per the MSLT II protocol [21]. The third iteration survey identified pervasive discordance between every 4 to 6 months, with 54% and 46% of respondents supporting its use every 4 or 6 months, respectively.

US was favoured as the imaging modality of choice for lymph node surveillance in view of the low positive predictive value of systemic imaging [8]. Moerhle et al. identified 100% sensitivity and 96% specificity rates based on the presence of any two malignant nodal characteristics on US among: hypoechoic centre, absence of hilar vascular pedicle, focal nodularity with increased vascularity and length to depth ratio less than 2 [22].

Much of this recommendation was derived from a discussion of the MSLT II trial data, in which patients with positive sentinel lymph nodes were randomized to completion

lymphadenectomy versus observation under an intensive US and clinical surveillance strategy. This trial identified no difference in MSS between groups at 3 years [21]. However, a significant improvement in locoregional control was afforded to the completion dissection group, as 92% of patients within this cohort were free from regional nodal basin relapse at 3 years, compared to 77% in the observation group [21]. This study highlighted a potential benefit of enhanced locoregional surveillance using a single imaging modality, without being powered to detect differences between the various imaging strategies.

The issue of concurrent regional nodal basin US utilization alongside PET-CT during the same 2-year intensive surveillance period was appropriately raised. Early retrospective radiologic studies on US and PET-CT revealed low sensitivity rates in the detection of subclinical nodal disease ranging from 8 to 24% and 10 to 21%, respectively [22–24]. A small retrospective study investigating the utility of high-resolution US (HRUS) and PET-CT compared to sentinel lymph node biopsy revealed HRUS correctly identified positive lymph nodes with greater accuracy (N = 2/17) compared to PET-CT alone (N = 0/17) [25]. Hence, the question of whether or not PET-CT is at least equivalent to US remains an active area of investigation.

### 7.5. How often Should Surveillance Be Performed?

High-risk surveillance should follow a 5-year schedule, beginning with an intensive 2-year period, followed by a less intensive 3-year period. This recommendation was achieved following a review of the available literature and responses from prior survey iterations, culminating in 85% agreement.

The most recent evidence centered on high-risk cutaneous melanoma is derived from clinical trials investigating novel immunomodulators in the setting of metastatic and unresectable disease. These trials focussed on therapeutic efficacy and survival as study endpoints. The corresponding surveillance strategies developed within these protocols have been loosely mirrored in clinical practice, despite the absence of clear supporting evidence as to its efficacy, as these studies were not designed with surveillance as a primary endpoint. What is known from earlier studies investigating intensive surveillance strategies is that the median time to relapse for Stage IIB to III disease was less than two years after treatment [3,7].

### 7.5.1. Frequency of Systemic Imaging

PET-CT was the preferred systemic imaging modality every 6 months during the first two years of HRS, followed by an annual schedule for a total of 5 years.

This recommendation was developed following a review of available data, limited to small prospective and retrospective series including heterogeneous approaches to identifying recurrences in resected melanoma [7,20,25]. There are no randomized studies investigating the benefits and varying frequency protocols of whole-body imaging. However, these early studies revealed that the majority of melanoma relapses for stage IIC and III disease occurred between 23 and 31 months [4,7]. The frequency of systemic imaging during this time varies widely across various reports ranging between every 3 and 12 months [4,7,13,21]. This suggests that a surveillance program with the intent of detecting asymptomatic relapses should be directed towards the first years after diagnosis and resection.

### 7.5.2. Frequency of Brain Imaging

Despite the paucity of evidence on both the diagnostic accuracy and frequency of CNS imaging for metastatic melanoma, annual brain MRI is appropriate for intensive and ongoing follow up.

A recent SEER report identified the stage-specific lifetime incidence proportions of cutaneous melanoma brain metastases to be 2.6% among patients who initially presented with localized disease [20]. This incidence increased significantly to 30.4% among patients who presented with distant disease [20]. Other series report the majority of brain metastases

develop within 24 months of treatment, with an incidence of 1.1% [3]. These results suggest CNS imaging may be beneficial in earlier stage disease than what is currently recommended by the NCCN guidelines (stage IIIC and IV).

### 7.5.3. Frequency of Nodal Surveillance

US of the draining nodal basin should be performed every 4 to 6 months during the first two years of intensive surveillance, followed by annual ultrasound for a total of 5 years.

In the absence of further data, safe follow up of patients with positive sentinel nodes without completion node dissection should be based on the MSLT II protocol—that is, regular US of the affected nodal basin to identify early recurrent disease as soon as possible [21]. To date, there are no head-to-head, comparative studies evaluating the efficacy of various US surveillance strategies for nodal melanoma recurrence. A recent Cochrane review included comparative accuracies of regional nodal basin US evaluations in the detection of clinically occult metastases prior to sentinel lymph node biopsy [26]. A summary of 11 US studies identified a cumulative sensitivity of 35.4% and specificity of 93.9% among 2600 patients [26]. In contrast, the estimated diagnostic sensitivity of PET-CT in the detection of nodal metastases for HRS ranged from 41% to 43%, and specificity ranged from 89% to 92% [27–29].

### 7.5.4. Frequency of Clinical Skin and Regional Nodal Basin Examinations

A full skin and regional nodal basin examinations should be performed every 6 months over the 5-year surveillance period.

Clinical follow up of melanoma patients has two purposes: to identify and treat recurrence and secondly, to identify a new melanoma primary. During a follow-up visit, patients undergo a verbal history and a full examination of the skin, lymph node basins and abdominal viscera. However, it remains to be determined whether this strategy leads to improved survival rates, especially in this era of systemic therapies for advanced stage disease.

The percentage of melanoma patients who develop a second primary melanoma (SPM) varies considerably in the literature, ranging from 2 to 20% [30–35]. This is likely attributed to epidemiologic differences in the incidence of melanoma across populations and to different study design measures. The highest incidence of SPM was reported in an Australian study of over 1000 patients, where 20% of patients developed a second melanoma over a median follow up of 16.5 years [35]. Other series from Europe and the United States have reported lower 5-year cumulative risks of developing SPM, ranging from 0.6%, to 11.4%, among cohorts including all stages of disease, including melanoma in situ [31–34]. A longitudinal study of 15 years showed a consistently low detection rate of SPM of 1% per year [36]. Cumulatively, these data suggest that a low but unwavering risk of SPM exists post-treatment that would support an endorsement for lifelong surveillance where resource availability permits.

Detection of SPMs have been shown to be thinner with a considerably greater proportion of in situ disease compared to first melanoma primaries [33,34,37,38]. Jones et al. identified better overall and melanoma-specific survival among patients who developed SPM. This is likely due to the fact that those patients who developed SPM shared favourable first primary melanoma characteristics conferring longer survival compared to non-SPM patients who did not live long enough to develop an SPM [34]. This suggests that the development of an SPM does not clearly negatively impact survival, and the characteristics of first melanoma primaries carry significance in guiding the frequency of clinical surveillance. Additionally, ancillary clinical detection tools such as total body photography and dermatoscopic documentation have been reported to aid in the detection of thinner and in situ SPM, demonstrating improved efficacy in clinical detection rates among dermatologists [33].

Cumulatively, these data demonstrate patients with cutaneous melanoma are at risk of developing SPM, hence close monitoring with clinical skin exams is critical to early

detection. Known clinical risk factors include male gender, fair skin, occupational UV exposure, high nevus count and familial melanoma risk and inability to tan [33–35,37]. These factors should be kept in consideration during clinical assessments. Due to the paucity of prospectively studied clinical surveillance strategies, the optimal frequency of clinical exams has yet to be determined. Hence, the expert panel agreed to biannual clinical skin and regional nodal basin examinations throughout the entire 5-year period for HRS.

## 8. Limitations

There were several limitations in the development of this recommendation statement. These included the non-standardized approach to expert recruitment throughout the three survey iterations, given the natural differences seen annually in the number of conference attendees. As a result, the number of expert participants varied between each survey iteration. Despite the inconsistency in the number of participants, the authors believe this natural variation brings a new perspective to each survey iteration, drawing novel ideas and considerations to the discussion during each phase, to reach at best, a consolidated majority opinion. Further, the authors acknowledge the recommendations offered here are based on limited data in the absence of prospective and randomized data targeting surveillance strategies.

## 9. Recommendations

The Canadian Melanoma Conference expert participants agreed upon the following high-risk surveillance schedule for treated, cutaneous melanoma based upon the most up to date and recent available data (Tables 1 and 2):

**Table 1.** Key recommendation points for HRS in cutaneous melanoma based on expert consensus from the 14th annual Canadian Melanoma Conference in 2020.

| Key Recommendation Points |
|---|
| The total duration of surveillance agreed upon is 5 years, consisting of biannual visits. The period of intensive surveillance consists of the first 2 years post-treatment. |
| High-risk cutaneous melanoma is defined as any patient with resected stage IIB to stage IV disease per the 8th edition of the AJCC staging system. |
| Clinic visitation including full examination of the skin, regional nodal basins, and abdominal viscera should follow a biannual schedule throughout the 5-year period. |
| Biannual PET-CT and annual brain MRI are the preferred imaging modalities for the detection of asymptomatic, systemic and central nervous system metastases, respectively. |
| There was no agreement on the frequency of regional nodal basin surveillance with US, hence the recommendation is left to the discretion of the provider. However, 4 to 6 months is suggested. |
| The final 5-year schedule on high-risk melanoma surveillance presented here was based on national expert agreement, recognizing regional resource limitations that may impact local practice guidelines. |

**Table 2.** The recommended 5-year surveillance schedule for high-risk melanoma patients.

| Evaluation | Period of Intensive Surveillance | | | | | | | | | 30 MO Visit 6 | Year 3 Visit 7 | 42 MO Visit 8 | Year 4 Visit 9 | 54 MO Visit 10 | Year 5 Visit 11 |
|---|---|---|---|---|---|---|---|---|---|---|---|---|---|---|---|
| | Postop + Visit 1 | 4 MO | 6 MO Visit 2 | 8 MO | Year 1 Visit 3 | 16 MO | 18 MO Visit 4 | 20 MO | Year 2 Visit 5 | | | | | | |
| Physical Exam | x | | x | | x | | x | | x | x | x | x | x | x | x |
| CT/PET or CT * | | | x | | x | | x | | x | | x | | x | | x |
| US ** | | x | +/− | x | +/− | x | +/− | x | +/− | x | x | x | x | x | x |
| MRI Brain | | | | | x | | | | x | | x | | x | | x |

\* Routine CT may replace PET CT where PET CT is not readily available. \*\* US of regional nodal basins is recommended every 4 to 6 months. Recommend a total of 11 clinic visits over 5 years.

**Supplementary Materials:** The following are available online at https://www.mdpi.com/article/10.3390/curroncol28030189/s1.

**Author Contributions:** Each author made substantial contributions to the conception and design of this work. All authors (C.W.L., J.G.M. and N.D.) participated in the acquisition, analysis, and interpretation of data, as well as the creation, drafting, and revision of this work. Each author has approved the submitted version and agrees to be personally accountable for their own contributions and for ensuring that questions related to the accuracy or integrity of any part of this work. All authors have read and agreed to the published version of the manuscript.

**Funding:** This research received no external funding.

**Institutional Review Board Statement:** Ethical review and approval were waived for this study, as this research was based on review of published/publicly reported literature.

**Informed Consent Statement:** Verbal informed consent was obtained from all subjects involved in the study.

**Data Availability Statement:** All data are presented in the study.

**Acknowledgments:** We thank the 2019 and 2020 CMC committee chairs for providing the opportunity and forum to present, discuss and disseminate knowledge in the development of this consensus document at the annual Canadian Melanoma Conferences in 2019 and 2020. The authors also thank the meeting participants, meeting sponsors and conference support staff for their contributions to the development of this consensus statement.

**Conflicts of Interest:** The authors declare no conflict of interest.

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
