# Peer review of "Canadian Melanoma Conference Recommendations on High-Risk Melanoma Surveillance: A Report from the 14th Annual Canadian Melanoma Conference; Banff, Alberta; 20–22 February 2020"

_curroncol, doi:10.3390/curroncol28030189_

Round 1

Reviewer 1 Report

On page 1 line 36-38, the authors failed to mention dermatologists and yet on page 5 line 158 they note dermatologists.   On page 10 line 349-350 the authors state "studies consistently report the greatest risk of SPM to occur within the first 2 years after treatment." This statement is incorrect.  The Kaiser Permanente study from 2015 showed a consistent rate of additional primary melanomas of 1 %/year over 15 years.  This speaks against the need for special focus on the first 2 years and more for lifelong follow-up if practical or feasible. 

Multiple primary melanomas among 16,570 patients with melanoma diagnosed at Kaiser Permanente Northern California, 1996 to 2011 Megan M. Moore, MD,c Alan C. Geller, RN, MPH,b E. Margaret Warton, MPH,a Joan Schwalbe, MS,a and Maryam M. Asgari, MD, MPHa J Am Acad Dermatol 2015;73:630-6. 

This is supported by the study they quote from Australia that highlighted second primary melanomas appearing over a span of almost 2 decades.   Authors should include this citatation   Second primary melanomas in a cohort of 977 melanoma patients within the first 5 years of monitoring Aimilios Lallas, MD, PhD, MSc,a Zoe Apalla, PhD,b Athanassios Kyrgidis, PhD,c Chryssoula Papageorgiou, MD,a Ioannis Boukovinas, MD,d Mattheos Bobos, PhD,e George Efthimiopoulos, MD,f Christina Nikolaidou, MD,g Andreas Moutsoudis, MD,a Theodosia Gkentsidi, MD,a Konstantinos Lallas, MD,a Elizabeth Lazaridou, PhD,h Elena Sotiriou, PhD,a Efstratios Vakirlis, PhD,a and Dimitrios Ioannides, PhDa   J Am Acad Dermatol 2020;82:398-406.     This study found that SPM 1-2.5%/yr over 5 years.
